# Impacts of hydropower on the habitat of jaguars and tigers

Ana Filipa Palmeirim[1,2] & Luke Gibson [1✉]

The rapid expansion of hydropower across tropical landscapes has caused extensive habitat loss and degradation, triggering biodiversity loss. Despite known risks to freshwater biodiversity, the flooding of terrestrial habitats caused by dam construction, and associated impacts on terrestrial biota, have been rarely considered. To help fill this knowledge gap, we quantified the habitat loss following inundation of hydropower reservoirs across the range of two iconic species, jaguars and tigers. To do so, we compiled existing and planned dams intersecting the distribution of these apex predators. We found 164 dams intersecting the jaguar range, in total flooding 25,397 km$^2$. For tigers, we identified 421 dams, amounting to 13,750 km$^2$. As hydropower infrastructure is projected to expand in the decades ahead, these values are expected to increase greatly, particularly within the distribution of jaguars where the number of dams will nearly quadruple (429 planned dams). Despite the relatively few dams (41) planned across the range of tigers, most will intersect priority conservation areas for this species. We recommend a more cautious pursuit of hydropower in topographically flat regions, to avoid extensive habitat flooding which has occurred in the Neotropics, and avoiding dam construction in priority conservation landscapes for tigers.

[1] School of Environmental Science and Engineering, Southern University of Science and Technology, Shenzhen, China. [2] Present address: CIBIO, Centro de Investigação em Biodiversidade e Recursos Genéticos, InBIO Laboratório Associado, Campus de Vairão, Universidade do Porto, 4485-661 Vairão, Portugal. ✉email: biodiversity@sustech.edu.cn

Hydropower development, aimed to accommodate rising global energy demands with minimal environmental costs, has become one of the major drivers of habitat loss, fragmentation, and degradation worldwide[1,2]. Currently, 3700 hydroelectric dams (>1 MW of installed capacity) are under development[3], many in tropical developing countries which sustain high levels of biodiversity[4]. Despite known risks to freshwater biodiversity[5], dam construction is often assumed to not meaningfully affect terrestrial biota[6]. Our understanding of the trade-off between hydroelectricity generation and biodiversity will be vital as many developing nations continue to expand hydropower infrastructure at the potential risk to natural capital.

Human land-use modifies the structure and composition of native ecosystems at varying scales and intensities, ranging from mild degradation (e.g., logged and secondary forests) to a virtual complete destruction (e.g., cattle pastures and tree plantations). In the case of hydropower, the area occupied by reservoirs becomes entirely unusable for terrestrial species, while the freshwater habitat becomes severely deteriorated for aquatic species[2,7]. This is particularly relevant in lowland tropical forests where, given the relatively flat topography, impoundment reservoirs tend to flood large areas[8,9]. Beyond the extent of the reservoir, surrounding areas also suffer from habitat loss, fragmentation, and degradation due to higher human accessibility[10,11]. Combined, effects on terrestrial species include both direct habitat loss due to flooding and declines in local density in the surrounding landscape[12–14]. Due to their low population densities and large area requirements[15], apex predators are expected to be particularly susceptible to habitat loss caused by hydropower infrastructure—both inside and outside the reservoir boundaries.

In this study, we considered the potential impacts of hydropower development on jaguars (*Panthera onca*) and tigers (*Panthera tigris*), which occupy the apex predator positions across the Neotropics and Paleotropics, respectively. Jaguars have suffered from population declines, and their distribution between Patagonia and the Southwestern USA has retracted by 50%, justifying their current designation as Near Threatened[16]. Once widely distributed across Asia, tigers have disappeared from >93% of their original range over the past century[17], and are now considered Endangered[18]. These iconic apex predators play a critical role in ecosystem functioning[19] and can also serve as umbrella species, enhancing the conservation of co-occurring species[20]. Currently, the total population size of jaguars (173,000 individuals[21]) is estimated to be ~50 times higher than that of tigers (3200–3500 individuals[22]). Despite considerable differences in their conservation status, both feline species face similar threats, primarily in the form of habitat loss and poaching[16,18]. In this context, hydropower expansion has been identified as a potential key driver of habitat loss, and thus a threat to both jaguars[23] and tigers[22], but the magnitude of this threat has not yet been examined.

Here, we quantify the contribution of existing and future hydropower development to the decline of jaguar and tiger habitat across their geographic ranges. We compiled existing and planned dams intersecting the ranges of both species and quantified the habitat area lost due to the flooding of impoundment reservoirs. We expected the habitat of tigers to have suffered greater losses given the longer history of hydropower in the region as well as overall extensive habitat loss across the Paleotropics[24]. On the contrary, due to comparatively aggressive development plans in Neotropical countries[4], we predicted that future hydropower growth will more strongly affect jaguar habitat. To compare the impacts of hydropower on these two species, we also estimated the total population size of each species potentially affected by habitat flooding, matching available species density values with reservoir area. Finally, we illustrate the trade-off between hydroelectricity generation and population decline for jaguars in Brazil, where we could obtain sufficient data on reservoir area and electricity generation for both existing and planned dams. Our overarching aims are to identify key threat areas for both species and to weigh the trade-off between energy development and biodiversity conservation.

## Results

**Current hydropower footprint.** We identified 164 hydropower dams overlapping the distribution of jaguars (0.2 dams/10,000 km$^2$; Fig. 1a) and 421 dams intersecting the range of tigers (4 dams/10,000 km$^2$; Fig. 1b). Of those, 282 dams intersect areas where tigers are resident, 90.7% of which are in India (Fig. 1c), and another 139 dams intersect areas where tigers are possibly extinct. Neotropical reservoirs were much larger (mean ± SD = 154.9 ± 513.6 km$^2$; max = 4437 km$^2$) compared to those in Asia (32.5 ± 99.7 km$^2$;

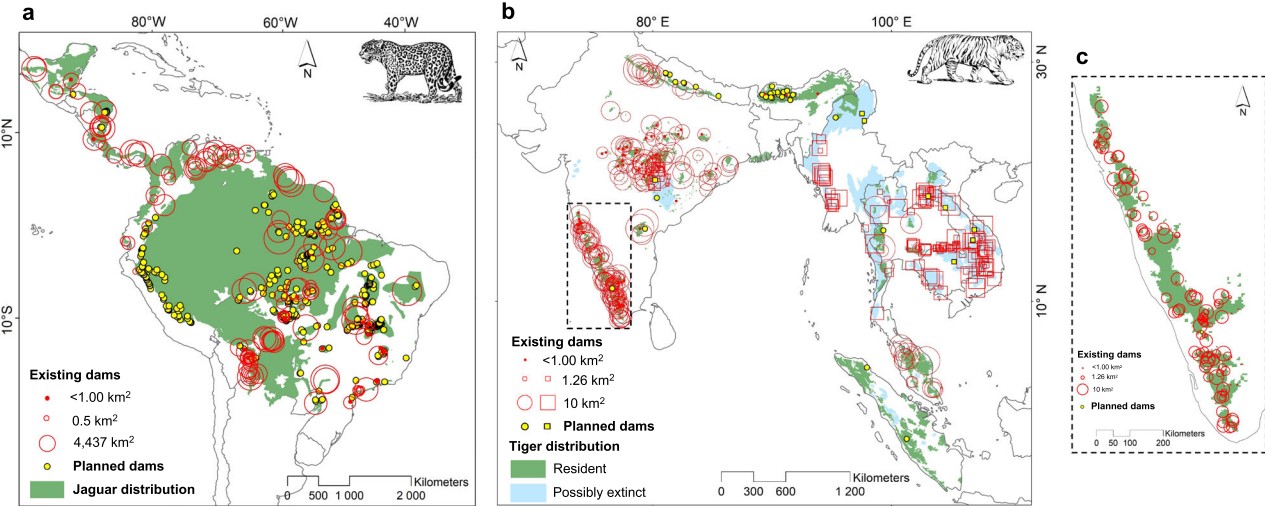

**Fig. 1 Distribution of existing and planned hydropower dams intersecting jaguar and tiger distributions. a** Hydropower dams intersecting the distribution of jaguars. **b** Dams across the distribution of tigers. Existing and planned dams are represented by red open circles and yellow dots, respectively, except in areas where tigers are possibly extinct, where dams are represented by *squares*. Circle/square size is proportional to reservoir area (log$_{10}$). **c** Inset showing Western Ghats in India.

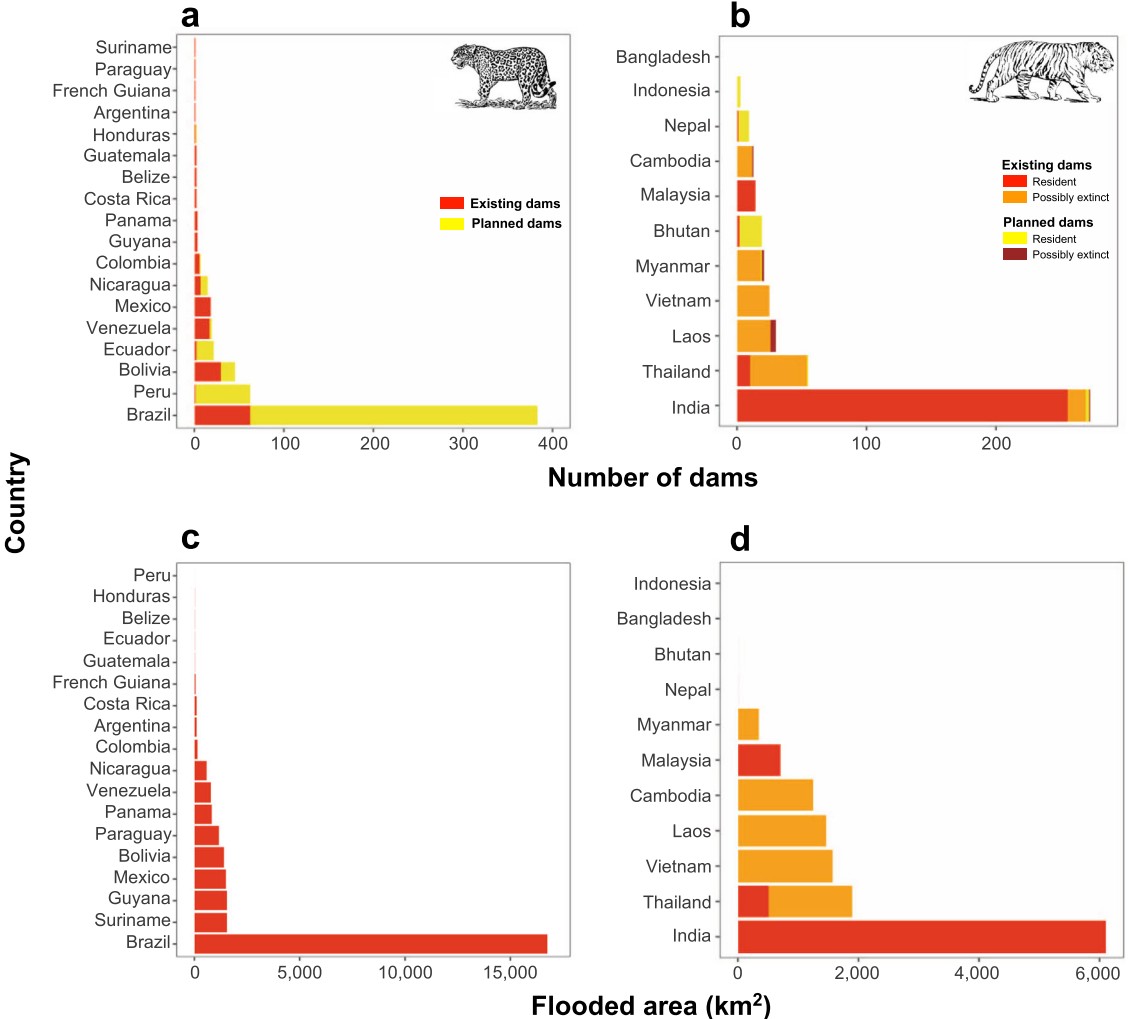

**Fig. 2 Number and flooded areas of existing and planned hydropower dams intersecting jaguar and tiger distributions by country. a** Number of existing and planned dams intersecting the jaguar distribution. **b** Number of existing and planned dams intersecting the tiger distribution. **c** Area flooded by hydropower reservoirs intersecting the jaguar distribution. **d** Area flooded by hydropower reservoirs intersecting the tiger distribution. Existing and planned dams are shown in red and yellow, respectively. For tigers, additional colors were used for dams located in areas where tigers are possibly extinct (existing: orange, planned: maroon). Panels **c** and **d** include only existing dams. Source data for all dams can be found in Supplementary Table 1.

max = 1198 km$^2$), leading to a total flooded area 1.8 times larger in jaguar habitat (25,397 km$^2$) than in tiger habitat (13,750 km$^2$; resident: 7611 km$^2$; possibly extinct: 6139 km$^2$). Given the larger amount of lost habitat, Neotropical dams potentially affected more jaguars, estimated as 915 individuals, corresponding to 0.53% of the total population. Asian dams, however, potentially affected a greater proportion of tigers, estimated as 729 individuals and corresponding to 20.8–22.8% of the total population (Supplementary Fig. 1).

**Planned hydropower expansion**. The future growth of hydropower will disproportionately affect jaguar habitat (Figs. 1 and 2a, b). We found >10 times more dams planned within the jaguar range ($n = 429$) compared to within the distribution of tigers (total: $n = 41$; resident: $n = 33$; possibly extinct: $n = 8$). Most will be constructed in the Amazon, the *Cerrado* dry forest hotspot (sensu ref. [25]) and the Andes-Amazon frontier (Fig. 1a). Brazil will be a major future source of hydropower, with 319 dams planned within the jaguar distribution. Within the tiger range, most planned dams will be located in areas where hydropower was previously absent or minimal, including Bhutan ($n = 17$) and Nepal ($n = 8$), or within priority areas for tiger conservation such as Sumatra ($n = 2$; Fig. 1b). Dam density is expected to increase three times over the jaguar range (0.6 dams/

10,000 km$^2$, considering existing and planned dams), but not substantially across the tiger range (4.3 dams/10,000 km$^2$).

**Trade-off: electricity generation vs. jaguars in Brazil**. The configuration of hydropower dams influences their impacts on apex predators, particularly due to differences in flooded areas and installed capacity, which are loosely correlated ($r = 0.40$, Supplementary Fig. 2). Dams sited in steeper slopes can produce high power without occupying large footprints, thereby having a comparatively smaller impact per unit electricity. We examined this trade-off for Brazil, where every 100 MW generation capacity of existing dams potentially affected a median of 0.54 jaguars (Fig. 3a); this ratio nearly doubled for planned dams, with a median of 0.97 individuals potentially affected per 100 MW (Fig. 3b).

**Discussion**
Although initially praised as clean green energy, hydropower development has become controversial due to its pervasive environmental impacts. Many studies have identified losses of both freshwater fauna induced by river disconnectivity[1,2] and terrestrial species assemblages due to habitat insularization often resulting from flooding[26,27]. Here we show that habitat loss in the

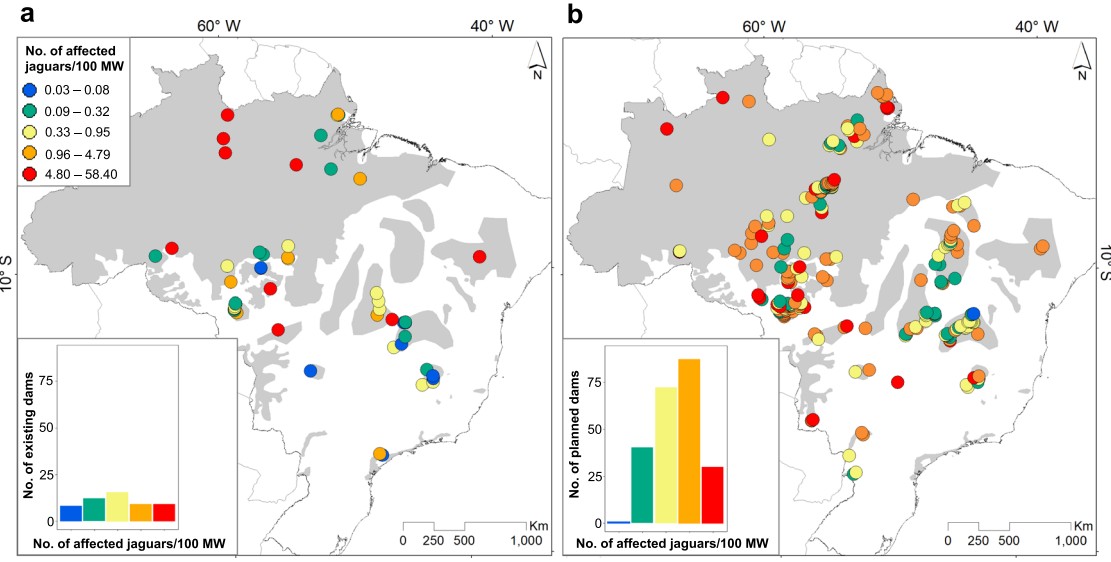

**Fig. 3 Distribution of existing and planned hydropower dams across the jaguar distribution in Brazil. a** Existing dams ($n = 53$). **b** Planned dams ($n = 230$). The distribution of jaguars in Brazil is shown in gray. Points are colored according to the ratio between the potential number of affected jaguar individuals and electricity generation capacity (100 MW) on a $\log_{10}$ scale (see detailed "Methods" section).

aftermath of hydropower development also affects terrestrial species, as illustrated for jaguars and tigers, with 0.3% (26,554 km²) and 0.7% (7304 km²) of their global distributions converted to reservoirs, respectively. In tropical lowlands, hydropower reservoirs typically extend over riparian habitats and floodplains, known to be key habitats for both species[28,29], particularly due to high prey availability[30]. In this sense, it is possible that the habitat flooded by reservoirs is of particularly high quality and importance for these predators.

Among the distribution of the two felids, habitat for jaguars has been affected by a lower number of hydropower dams. However, the area flooded by dams in the Neotropics was much larger, potentially also affecting a higher number of jaguars, which would still represent a smaller fraction of the total population size of this species. In the near future, we can expect considerable further losses in the habitat of jaguars, given the elevated number of planned dams in this region. Also, as the amount of energy produced per flooded area is a function of topography, hydropower development in relatively flat lowland forests creates not only larger reservoirs, but also less efficient dams[9]. Our results show that future dams intersecting the jaguar range, at least in Brazil, will flood increasingly larger areas for the same amount of hydroelectricity produced. This is illustrated by the dramatic 650% increase in the number of dams with the worst trade-off between electricity generation and number of jaguars potentially affected (Fig. 3). As hydropower efficiency decreases, the trade-off between electricity generation and ecological impacts will only deteriorate, contributing towards more habitat loss and elevated threats to biodiversity for each megawatt generated.

Regarding the scenario for tigers, an endangered species with a long history of hydropower development inside its distribution, so far, flooded areas hypothetically affected >20% of the global population of this species. Hydropower has thereby become an important driver of tiger habitat loss. Despite the relatively lower number of dams planned across its range ($n = 41$), tiger persistence does not appear to be properly considered in future hydropower development within the region. Indeed, most planned dams overlap important priority tiger landscapes as well as protected areas or complexes (e.g. Nepal, Bhutan, and North Sumatra)[31] (Fig. 1b). In particular, two dams are planned for construction in Sumatra near the Leuser Ecosystem, home to an important source population of Sumatran tigers, a critically endangered subspecies[32]. Such future projects have the potential to derail the St. Petersburg Declaration on Tiger Conservation in accomplishing the lofty goal of doubling the global population of this species (Saint Petersburg, Russia, November 23, 2010)[33,34].

While our study quantified habitat loss due to flooding following river damming across jaguar and tiger ranges, there are other detrimental impacts caused by hydropower development. First, hydropower reservoirs are increasingly located in remote areas, and their construction greatly increases human access to these frontier wilderness areas (e.g., construction of roads and transmission lines[35]). Construction of such infrastructure contributes towards the additional loss, fragmentation, and degradation of the habitat surrounding reservoirs[10,11]. This further reduces the potential of these areas to support viable populations of jaguars[36,37] or tigers[32,38], and may eventually disrupt metapopulation dynamics[39]. Second, damming in relatively small forest areas already harboring reduced populations of top predators is expected to have further implications, potentially precipitating their local extinction[40]. This might be the case for some populations of jaguars in the Atlantic Forest and Pantanal of Brazil, and for tigers in Central India (see Fig. 1). On the other hand, displaced individuals might move to habitat areas surrounding reservoirs, eventually increasing species density therein if a suitable prey baseline is available[41], there is minimal hunting pressure, and the appropriate spatial requirements are met[42]. In light of evidence of habitat degradation in the aftermath of damming[10,11,35] and the unsuccessful relocation of individuals occupying habitats on the verge of damming by rescue operations[43], we consider such an increase to be unlikely. For instance, one population of marsh deer (*Blastocerus dichotomus*) in the Brazilian Pantanal declined by 54% after damming due to habitat reduction and deterioration of food availability[13]. Admittedly, our estimates on the number of jaguars and tigers potentially affected might be an overestimate, if animals can persist in nearby non-flooded habitat, or an underestimate, given that dam construction is often associated with deforestation and further habitat loss in surrounding areas. Unfortunately, to date, no study has evaluated the in situ impacts of reservoir filling for either of these felid populations; this baseline information should be considered essential and a target for future studies.

Apex predators play a crucial role in ecosystem functioning and the delivery of ecosystem services (e.g., carbon sequestration, fire and the regulation of disease and invasive species)[44]. For example, jaguars and tigers both exert top-down control of lower trophic levels[19,45], preventing the irruption of herbivores which could impede forest regeneration, culminating in an "ecological meltdown"[26]. Both species further serve additional vital roles in the countries where they are found, as flagship species attracting ecotourists, and as umbrella species supporting critical ecosystem services[44]. Although jaguars and tigers are primarily affected by habitat loss and poaching[16,18], here we show that hydropower development constitutes an important driver of such habitat loss. This elevates the overall importance of preserving terrestrial habitats required to sustain populations of apex predators. In fact, even semi-aquatic apex predators decrease in abundance in the aftermath of damming due to the poor habitat quality offered by reservoirs[7]. Our results suggest that the economic benefits of hydroelectricity generation do not always compensate for the negative environmental impacts, as already demonstrated for multiple hydropower reservoirs in the Brazilian Amazon[46,47]. This issue is particularly relevant for developing countries that still harbor high levels of biodiversity, and on which payment of ecosystem services has the potential to alleviate poverty[48].

We finally highlight strategies which could help mitigate the impacts of hydropower infrastructure. For existing reservoirs, surrounding habitats should be included in protected area systems to avoid expanding the footprint of hydropower and triggering the decline of top predators, overall biodiversity, and associated ecosystem services. This proposed measure is compatible with those recommended by other studies considering the effects of human disturbance on jaguars[49] and tigers[34]. Yet, given that hydropower reservoirs often facilitate human access to formerly remote frontier areas, appropriate enforcement efforts must be allocated to protected areas[50], including but not limited to tiger priority landscapes[51].

Looking to the future, planned hydropower projects should minimize the trade-off between biodiversity loss and electricity generation, most easily achieved by avoiding development in topographically flat regions, especially important for jaguars in the Amazon basin. For tigers, an endangered species found in relatively small (<10,000 km$^2$) remnant habitat patches, any planned dams intersecting priority tiger conservation landscapes (sensu ref. [52]) should be aborted. Considering the potential of hydropower to meet future energy demands, we recommend a more cautious balance between electricity generation and the conservation of terrestrial habitats, a key ingredient towards sustainability.

To achieve such a balance, strategic planning and environmental impacts assessments must be carried out with the inclusion of experts who can assess the potential ecological impacts of proposed hydroelectric projects. Indeed, such assessments should provide adequate technical information to increase the influence on policy decisions[53]. Accounting for such recommendations within country-level legislation would be a major policy challenge preventing further reduction of jaguar and tiger habitat across their ranges, while also maximizing the potential of these species' long-term persistence and ensuring adequate energy production. Given the crucial roles of apex predators, accounting for the impacts of hydropower development on these species will help avert regional scale biodiversity collapse and associated losses of ecosystem services.

## Methods

**Data acquisition**. We exhaustively searched for databases, published studies, and reports including information on either existing or planned dams located in the current range states hosting jaguar (i.e. Argentina, Belize, Bolivia, Brazil, Colombia, Costa Rica, Ecuador, French Guiana, Guatemala, Guyana, Honduras, Mexico, Nicaragua, Panama, Paraguay, Peru, Suriname, and Venezuela) and tiger populations (i.e. Bangladesh, Bhutan, Cambodia, India, Indonesia, Laos, Malaysia, Myanmar, Nepal, Thailand, and Vietnam). We searched Web of Science and Google Scholar using the following keywords: hydroelectric dams, [country name] [dam status: in operation/under construction/planned]. For those countries with sparse information on planned dams (e.g., Nicaragua, Honduras, Thailand, and Malaysia), we searched using the same keywords translated into the local languages (Spanish, Thai, and Malay). Whenever geographic coordinates were not available, we obtained location information by searching for the respective dam name on Google or Google Earth. For each dam, we collected information on location (geographic coordinates), status (existing or planned), reservoir area (km$^2$), and installed capacity (MW). Whenever reservoir area was not available for existing dams, we manually measured it using Google Earth Pro. Reservoirs less than 0.01 km$^2$ were considered to not meaningfully affect the home range of jaguars (e.g., 13.4–2914.9 km$^2$ [ref. [54]]) or tigers (397 km$^2$ [ref. [32]]) and were not included in further analyses. Dams were classified as (1) existing, if already in operation or under construction with known reservoir area; and (2) planned, if its construction had not yet begun (including both dams with and without studies/licensing completed), or if its construction had begun but information on reservoir area was not available (suggesting its preliminary state of construction).

For tigers, we provide estimates on habitat loss considering both areas where tigers are resident and where this species is possibly extinct. We excluded the remnant tiger populations occurring in Russia and China due to the very low levels of hydropower development across that part of the tiger range (i.e., only one reservoir was identified in China, occupying just 5.1 km$^2$ of the current tiger range[55]). In addition, this region is outside the tropics, the primary target for future hydropower development[4] and also the focus of this study; hydropower development is not expected to form a major threat to tigers in this part of their range.

### Statistics and reproducibility

*Measurements of habitat loss due to flooding.* After cataloguing all dam information, we used the geographic coordinates provided by the source to overlap with the IUCN distribution of jaguars[16] and tigers[18] (Supplementary Fig. 3). We then summed the area of existing reservoirs within the species range.

To evaluate the potential impacts of existing hydropower on predator population size, for each existing dam, we first estimated the potential number of jaguar/tiger individuals affected by habitat flooding. To do so, we matched the area of each existing reservoir with the nearest available estimate of species density to obtain the potential number of affected individuals. For tigers, we compiled species densities from primary and gray literature within the geographic range where tigers are considered both resident and possibly extinct[18]. For jaguars, we considered densities reported within the studies compiled in a recent study[21], except for Honduras, Nicaragua, and Guatemala, where there were no available density estimates. For reservoirs located in those countries, we used density values generated at the country level[21] (density estimates and information on the study sites where densities were obtained can be found in Supplementary Data 1 and 2). We then summed all individuals potentially affected at each existing reservoir and related that to estimates of total population size. We considered the total population of tigers to range between 3200 and 3500 individuals[22] and of jaguars to be 173,000 individuals[21]. Given evidence on animals rescued from flooded areas and released in habitat surrounding the reservoir (see ref. [43] for a recent review), here we assumed that the predators would not be likely to survive the habitat loss resulting from reservoir flooding; even if displaced to surrounding intact habitats, the available prey base, habitat area, and potential resulting competition would likely cause higher mortality and thereby maintain the estimated densities[56,57]. While this study aims to illustrate and compare the different scenarios for jaguars and tigers under existing and future hydropower development, we acknowledge that these are rough estimates that were not based on in situ studies of the response of these species to habitat flooding, which are currently unavailable. We therefore urge caution when interpreting these results.

*Trade-off between electricity generation and jaguar population decline.* To determine the ratio of the number of individuals affected per unit of electricity generated (100 MW) by existing and planned dams, we used data on the installed capacity and reservoir area of both existing and planned dams. We carried out this analysis only for jaguars in Brazil because data on installed capacity and reservoir area for both existing and planned dams were only available for Brazil, where more than half of the total jaguar population remains (approx. 86,800 individuals[21]). Dams intersecting areas with less than 0.0001 jaguars km$^{-2}$ were considered to not meaningfully affect jaguar habitat, and thus not considered; from a total of 294 dams, we selected 283 dams for this analysis (Supplementary Data 3). Here we aimed to provide a comparison of the energy produced per area flooded between existing and planned dams. Again, given the uncertainty in the number of jaguars affected by each reservoir, our estimates are rough and we urge caution when interpreting these results. We further investigated how reservoir area correlated with installed capacity of existing and planned reservoirs intersecting the jaguar distribution in Brazil, using a Pearson correlation.

**Reporting summary**. Further information on research design is available in the Nature Research Reporting Summary linked to this article.

## Data availability
The datasets generated during the current study are available in the Supplementary Data 1 to 3.

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

## Acknowledgements

We thank G.R. Clements for assistance in the compilation of data on tiger densities, and P. Fearnside and C. Wight for useful advice to identify dams intersecting the distribution of jaguars. AFP was supported by the Outstanding Postdoctoral Fellowship of the Southern University of Science and Technology (SUSTech), and is currently funded by the European Union's Horizon 2020 research and innovation programme under grant agreement No. 854248. L.G. was supported by the China Thousand Young Talents Program (K18291101), as a Guangdong Government distinguished expert (K20293101), and by the Shenzhen Government (Y01296116).

## Author contributions

L.G. conceptualized the idea, A.F.P. performed data analysis and wrote the original draft, and both authors revised the manuscript.

## Competing interests

The authors declare no competing interests.
