## [Peer Review File · Communications Biology]

Reviewers' comments:

Reviewer #1 (Remarks to the Author):

Brief summary of the manuscript

Palmeirim and Gibson evaluate the potential habitat loss for tigers and jaguars associated with reservoir flooding from future hydropower operations using existing datasets from multiple sources.

Overall impression of the work

The authors use a simple approach involving the assessment of overlapping areas between hydropower influence and published animal ranges and densities. This is a very interesting and simple idea that sheds light on the complex trade-off between conservation and energy needs that many developing countries face worldwide. Yet, there are some concerns of using this approach to support the management actions that the authors recommend such as not constructing planned dams in some regions. The reduction in habitat size due to reservoir flooding may not result in changes of population sizes. For example, in areas where abundances are already low and thus, these systems would be able to support higher animal densities. Considering the last, the assumption of tigers and jaguars not surviving the habitat loss from reservoir flooding even if displaced to surrounding intact habitats is questionable.

Specific comments and recommendations

-The authors need to clarify that deforestation and poaching are the main drivers of the reported declines in tiger and jaguar populations and thus, hydropower generation represents an additional stressor for habitat loss rather than a main driver as could be interpreted from reading this article. This aspect should be addressed in the introduction and discussion.

-The article would benefit from including a thoughtful discussion about the role of conservation of terrestrial habitats versus hydropower generation. This will be relevant for planning strategies in developing countries. These aspects could be considered in future management decisions about new hydropower projects. For example, it would be great to contrast the benefits of energy generation and the ecosystems services provided by the conservation of biodiversity.

-In the methods section, the authors should provide additional information about the quality of datasets used for tiger and jaguar densities, especially the datasets compiled from gray literature (e.g., list the data sources and provide an assessment of their credibility). Also, a more detailed explanation is needed about how published estimates of species densities from nearby regions are used. Is any aspect of differential habitat conditions among these regions considered? Similarly, the authors need to provide the sources of information related to the list of dams (i.e., Table S1 and S2). Lastly, as I mentioned above, the assumption that tigers and jaguars would not survive the habitat loss from reservoir flooding is weak. A comprehensive analysis of the literature could resolve this along with an evaluation of evidence that the expected elevated competition under habitat reduction is feasible. However, a special consideration should be taken for populations that are already in decline due to the carrying capacity of these systems are arguably below their historical conditions so higher densities might not be an issue.

Reviewer #2 (Remarks to the Author):

This is an interesting topic that perhaps does not receive the attention that it may deserve, although I feel that the manuscript lacks detail and depth of its analysis and interpretation of results. As I lay out in more detail, I have some caveats on omissions in the data and focus of the analysis, as well as too strong of a focus on extrapolating population effects rather than discussing habitat loss.

Line 44: The range in home range size is understated as it can be much larger for jaguars, see Morato et al. 2016 (PLoSone), McBride and Thompson 2018 (Mammalia), and Thompson et al. 2021 (Current Biology) for estimates of jaguar home range and Morato et al. 2018 (Biological Conservation) and Thompson et al. 2021 for importance of forest cover for jaguars.

Lines 46-49: I think there should be a greater emphasis on the habitat loss and less on the extrapolation of population change since that is considerably more conjectural.

Results: It would be helpful to have a summary table by country of number of planned projects and their status within the main text. Based upon the methods this was done but where are the results? There are no dam projects planned in tiger habitat or potential habitat in Russia or China? I think that the effects of dams on unoccupied, potential tiger habitat needs to be included. Shifting the emphasis on to habitat loss would facilitate this. I see in the methods that these areas were excluded. This is a major flaw in the analysis as it overlooks small populations or unoccupied habitat that are all important, especially for tigers. The supplement of the dam projects, needs to include an estimate of habitat loss by the construction of each. Why are just jaguars in the table and not tigers? The graphs in the supplemental material do not really convey the information well. Also, why are their graphs and results related to jaguars and damns and not for tigers? What are the results for tigers?

Although the analysis addresses habitat loss due to flooding, there are two aspects of this that are not addressed. Hydroelectric projects are generally accompanied by protected areas around their perimeters to avoid siltation. It could be argued that damns might result in a net gain of protected habitat. I am not saying that that is the case but it needs to be evaluated. This is also related to what habitat is lost, riparian habitats and flood plains are preferred habitats for tigers and jaguars, a better discussion of habitat quality in relation to loss is needed.

Throughout the results and discussion, the authors talk in absolutes of loss of individuals and this is not the case, they are estimates and should be treated as such. Also, density estimates common with estimates of variance, the estimates of losses from damns should incorporate that uncertainty.

Additional effects of population and infrastructure (roads) growth associated with dams needs to be discussed. Infrastructure that allows human access negatively impacts jaguars and tigers. For example, see Carter et al. 2020 (Science Advances), Espinosa et al. 2018 (PLoSone), Thompson et al. 2021, Kerley et al. 2002 (Conservation Biology).

Line 107: Again, talking about % habitat loss is more useful than extrapolating loss of individuals.

Line 115: Although very dependent upon accurate density estimates I think this metric of individuals/kw is interesting as it puts the effects of dams into a common currency. It should be used more. Also, per project and per country reporting in the supplemental material would be useful, and a summary within the main text as well.

Line 119-130: For conclusions and applications this section is weak. The implications and applications of the results need to be better framed, including what can be gained from accounting for dam effects with the conservation of these species.

Lines 154-165: Start a new paragraph after line 165. This whole paragraph needs to be more detailed and clearer. It is not entirely clear how the data were organized and the analysis undertaken.

Line 165: Starting here an analysis of habitat loss should be included. The extrapolation to population numbers is useful for discussion but prone to a lot of uncertainty (uncertainty in density estimates, fate of displaced individuals, potential protection of buffer areas, etc.).

Lines 176-177: "Only dams intersecting areas with a minimum of 0.0001 jaguars km⁻² were selected (283 dams)." What is this based upon? Why a value with so many significant digits? What is the justification for this cutoff?

Lines 178-179: "We further analyzed how installed capacity of those dams predicted the loss of jaguars using a Generalized Linear Model." Why just for jaguars? This analysis is not discussed much in the text. What was tested exactly?

Reviewer #3 (Remarks to the Author):

Thank you for your submission to Nature Communications. I find your manuscript to be well written and of a topic that is most certainly under appreciated. My main comments relate to improving the framing of the problem. For instance, your Introduction does not currently provide much information about the adverse effects resulting from hydropower. This seems to be left for the reader to implicitly understand. A short sentence or two to highlight the main impacts of hydropower would help to guide the reader on the extent of the problem. Secondly (and related), it is unclear how big a problem hydropower actually is from your Introduction (with more significant effects on tiger populations than (larger) jaguar populations), other than noting that hydropower will increase by >30% over the coming decades. Infrastructure development projects, for example, are expected to add 3 to 5 million km of road across ecosystem globally over the next 50 years (Meijer et al, 2018), with potentially dramatic consequences on ecosystem structure and function. How do impacts from hydropower compare? This is not to say that one type of land-cover change is more important to highlight than another, but it is to say that I believe a short section is needed to contextualize the rapid changes that are occurring across ecosystem globally. I consider land-cover change to be one of the greatest threats to terrestrial biodiversity. But, animals, such as jaguar still move through disturbed landscapes (see Morato et al. 2016 and 2018), up to some (although unknown) threshold of human disturbance. Dam construction and its associated conversion of terrestrial to aquatic habitat will surely make areas unusable, but what level of mortality will be caused by the loss in area and assumed decrease in landscape carrying capacity? The point here is to be careful about assumed responses or at least, to tender these consequences appropriately.

Your title does not fit the context of the manuscript. Other than the inference that current and future hydropower dams will decrease the habitat area and result in increased mortality, there is no discussion on impacts or responses of jaguar or tiger to anthropogenic factors. While this is not the focus of your analysis, what lessons are we to take from your analysis? That species area will be decreased?

Further, why are jaguar and tigers chosen for the analysis? I have no problem with doing so, but some discussion of why these apex predators were chosen would be appropriate. Is it because they have large area requirements? Is it because they are sensitive to anthropogenic disturbance? Some information about the importance of apex predators would seem necessary to make your case for the importance of your findings or that these species act as umbrella species for a variety of biodiversity that will also be lost with increases in hydropower.

Lastly, I noted that home range estimates provided in your Introduction range from 40-400 km². Morato et al. 2016, however, publish findings on the home range of jaguar using the Continuous Time Movement Modeling framework to more appropriately incorporate the autocorrelation structure inherent in most modern-day tracking datasets. Findings from this analysis illustrate that CTMM derived home ranges are 1-5 times larger than estimates based on previous methods, an easy update to make. I know your analysis is not based on home range estimates, but are there inaccuracies in IUCN distributions or density estimates that could result in inaccuracies of your conclusions? In the end, it would seem that your results provide a best guess approximation based on relevant assumptions, but errors do propagate and are worth further discussing.

I hope these comments are helpful and constructive in providing information towards an improved manuscript.

Reference:

Meijer, J. R., Huijbregts, M. A. J., Schotten, K. C. G. J. & Schipper, A. M. 2018. Global patterns of current and future road infrastructure. *Environ. Res. Lett.* 13.

Morato et al. 2016. Space Use and Movement of a Neotropical Top Predator: The Endangered Jaguar. *PLoS One* 11, 1-17.

Morato et al. 2018. Resource selection in an apex predator and variation in response to local landscape characteristics. *Biol. Conserv.* 228, 233–240.

Specific comments throughout the manuscript:

Abstract:

Hydropower is certainly a primary threat to freshwater biodiversity. But, I believe you should be broadening this statement considerably to lay the foundation for your manuscript. What about the catastrophic collapse on anadromous fish populations, like salmonids, where the impacts of hydropower extend well beyond freshwater systems? I would broaden this statement to showcase these immense problems. This doesn't need to be an extensive addition. You are correct that the 'primary' threat is freshwater biodiversity, but perhaps broadening this statement would better showcase the problem.

Introduction:

I think there is a need in the Introduction to focus greater attention on the impacts of hydropower and how systems have changed (reference to published sources), in an effort to guide the reader. Currently, almost no attention is provided on this aspect. How does hydropower construction compare with other forms of infrastructure development (i.e., roads, railways, pipelines) that have contributed to biodiversity decline and are increasing rapidly across terrestrial ecosystems globally. Most certainly, land-cover change is one of the greatest threats to biodiversity globally. But, how much impact does hydropower have? Perhaps this is one of the main goals of your study, with emphasis that hydropower has a much more significant effect than previous thought. This should be made clear to the reader.

Line 44 - As noted above, do you have a citation for the home ranges listed. Morato et al. 2016 calculate home ranges for jaguar across various biomes in South America using the Continuous Time Movement Modeling framework, a method that incorporates the autocorrelation structure inherent in most modern-day GPS tracking datasets. Noonan et al. (2019) - A comprehensive analysis of autocorrelation and bias in home range estimation - provide an analysis of how this method compares with other home ranges and find that previous methods underestimate home range areas. In the Morato et al. (2016) study, jaguar home ranges varied from 24 km² to 1268 km², representing a 1 - 4.8 increase in home range area from previous methods (same data, new method). Since your analysis is based specifically on distribution areas (not necessarily home ranges), you could be drastically overestimating how many individuals will be impacted by dam construction.

Results:

Line 62: You mention that "tigers have been hit hardest by past hydropower." Do you have a reference for this statement? One could easily argue that land-cover change resulting from high human population density is the main reason why so little habitat remains for tiger. Dams are another/next stage of land-cover change with potential additional negative effects because the "land" becomes unusable for the species.

Line 63: "> Ten times more dams planned in jaguar habitat." Yes, but the area is much larger. What's the percentage or density of dams on the landscape?

Line 65: The 'hotspot' cerrado. What do you mean 'hotspot'? Are you referencing the importance, biologically, of this habitat that is being rapidly lost and degraded?

Line 71: I suppose the question is whether establishing a protected area actually supports/protects these species. Far too often, protected area boundaries are altered or the protection status is changed altogether as a result of political will. Good examples exist on this topic in Kenya (i.e., Nairobi National Park boundaries altered to accommodate road construction) and Niger (i.e., the protection status of the Termit Tin Toumma Reserve - the largest terrestrial protected area in Africa - was changed to allow for oil/gas exploration). Even in the United States, various plans exist to allow for mineral extraction on protected lands. It begs the question as to the value of protecting areas if we (as a people) are not going to honor these agreements. Perhaps this is also something for the Discussion on recommendations.

Discussion:

Line 109: "In the near future, we can expect considerable further losses of jaguars, given the elevated number of planned dams across this species range." Similar to comments made in your Methods, is there published research that you could reference on the response of either species (or related) to land-cover change/development. Surely with vast swaths of the Amazon being deforested and converted to agriculture, there must be research into how jaguars respond to these changes. Perhaps dam construction is even more deleterious because it renders the habitat unusable when terrestrial habitats are converted to aquatic systems.

Last paragraph/recommendations section: I think you should specifically add that scientists/conservationists need to be included in the Discussion on the potential impact of these proposed activities to provide a point-of-view about potential impact, through Environmental Impact Assessments or other, prior to actual construction.

Methods:

Line 155: "with the IUCN⁷ and jaguars⁸". What does that mean? Do you mean the distribution map provided by the IUCN on tiger and jaguar?

Line 161: You provide an example of local species densities (e.g. 10)? Why are you not specifically listing your data sources? Are they too numerous to list? If so, these still should be listed in your Supplemental Materials documentation, as the data incorporated have the potential to bias your results (both positively and negatively).

Line 170: "We assumed that the predators would not survive the habitat loss resulting from reservoir flooding; even if displaced to surrounding intact habitats, the resulting elevated competition would likely cause higher mortality and thereby maintain the estimated densities." While I agree with this assumption, is there published literature on how each species respond to habitat loss/displacement? All of your analyses are based on this assumption (loss in habitat from hydropower construction will result in a decrease in landscape carrying capacity and population decline). Important that what you are actually measuring is potential habitat loss.

Figures:

Fig 2. Insets c, d, and e provide little additional information to the reader. I would remove. These insets (c, d, e) are also listed in Figure 2b as 'a', 'b', and 'c' (not c,d,e)

Fig S1. Tiger distribution (S1a) is shown, prior to jaguar distribution (S1b). In the main text, however, jaguar (1a) precede tiger (1b). It doesn't matter which is listed first, but these should be consistent. In Fig S2, tigers are again listed prior to jaguar, but in Fig S3, jaguar are once again listed first.

Fig S3. The text should read "Number of existing (a,b) and planned (c,d) hydropower reservoirs intersection tiger (a,c) and jaguar (b,d) distributions, by country." The current way figure numbers are listed is confusing/hard to read (e.g., a,b existing).

Reviewers' comments:

Reviewer #1 (Remarks to the Author):

Brief summary of the manuscript

Palmeirim and Gibson evaluate the potential habitat loss for tigers and jaguars associated with reservoir flooding from future hydropower operations using existing datasets from multiple sources.

Response – Thank you very much for your positive comments and the time invested in reviewing our manuscript. In this revised version of the manuscript, we have addressed all points raised. Below, we detail how we addressed each point raised and indicate the corresponding lines of any changes in the main text.

Overall impression of the work

The authors use a simple approach involving the assessment of overlapping areas between hydropower influence and published animal ranges and densities. This is a very interesting and simple idea that sheds light on the complex trade-off between conservation and energy needs that many developing countries face worldwide. Yet, there are some concerns of using this approach to support the management actions that the authors recommend such as not constructing planned dams in some regions. The reduction in habitat size due to reservoir flooding may not result in changes of population sizes. For example, in areas where abundances are already low and thus, these systems would be able to support higher animal densities. Considering the last, the assumption of tigers and jaguars not surviving the habitat loss from reservoir flooding even if displaced to surrounding intact habitats is questionable.

Response – This is an important point and something we have thoroughly considered. We agree that it is possible that some or all of the jaguar and tiger individuals occupying the reservoir areas prior to river damming might survive in the aftermath of damming, being displaced to the habitat surrounding the reservoirs. In theory, in case the jaguar/tiger density is low in the reservoir surroundings (in the pre-filling scenario), displaced individuals might contribute to increase that density. That would be likely to happen if the carrying capacity of that habitat allows for that (e.g., Ramesh, K. et al. Status of tiger and prey species in Panna Tiger Reserve, Madhya Pradesh: Capture-recapture and distance sampling estimates.

Technical Report, Wildlife Institute of India, Dehradun, and Panna Tiger Reserve, Madhya Pradesh 2013). Yet, areas affected by hydropower development are often further degraded by

other human activities (e.g., construction of roads and expansion of urban and agricultural areas; e.g., Chen, G. et al. Spatiotemporal patterns of tropical deforestation and forest degradation in response to the operation of the Tucuruí hydroelectric dam in the Amazon basin. *Appl. Geogr.* 63, 1–8, 2015; Hyde, J. L. et al. Transmission lines are an under-acknowledged conservation threat to the Brazilian Amazon. *Biol. Conserv.* 228, 343–356, 2018), including increased human access to previously remote forest areas. For this reason, we believe that our estimate of the number of individuals affected by direct habitat loss due to the development of hydropower reservoirs might be an underestimate rather than an overestimate. Although evidence on animals rescued from flooded areas and then released in nearby habitat shows that most individuals fail to survive mostly due to high competition (see Alho, C. J. (2020) Hydropower dams and reservoirs and their impacts on Brazil's biodiversity and natural habitats: a review. *N.a. J. Adv. Res. Rev.* 6, 205–215, for a recent review), we currently lack any study reporting the effects of habitat flooding *in situ* on either jaguars or tigers. This limitation impedes us from drawing more precise conclusions on this matter. As an attempt to overcome this issue, following the suggestion provided by Reviewer #2, we have now changed the focus of our manuscript to emphasize habitat loss due to flooding rather than our estimates of the potential number of individuals affected. Also, due to such uncertainty, we have also changed the previous terminology of 'potential individuals lost' to 'potential individuals affected' throughout the text. Moreover, we have included a detailed explanation on the potential bias inherent to our estimates regarding the potential number of individuals affected. Indeed, it is possible that our estimates correspond either to an overestimate, in the case that animals can persist in the habitat surrounding reservoirs, or an underestimate, given that dam construction is often associated with further human disturbance that contributes to the loss and degradation of habitat in the reservoir surroundings (lines 178 – 181). Finally, we highlight that the overall purpose of our estimates on the potential number of affected jaguars and tigers is to obtain the proportion of the total population size that has already been affected by hydropower for each species. This allows us to illustrate the comparison between the contrasting scenarios of the two predator species: jaguars, which are Near Threatened, and tigers, classified as Endangered. This also allows us to compare hydropower development in the two regions where these species are distributed: the Neotropics, with a more recent history of hydropower development, and the Paleotropics, with an older history of hydropower development (lines 279 – 282).

Specific comments and recommendations

-The authors need to clarify that deforestation and poaching are the main drivers of the reported declines in tiger and jaguar populations and thus, hydropower generation represents an additional stressor for habitat loss rather than a main driver as could be interpreted from reading this article. This aspect should be addressed in the introduction and discussion.

Response – We agree and have followed this suggestion to address that both deforestation and poaching are the two main threats to both tiger and jaguar populations, whereas hydropower represents a driver of the overall habitat loss. This comment has been now addressed in both the Introduction (lines 66 – 67) and Discussion (line 190 – 191).

-The article would benefit from including a thoughtful discussion about the role of conservation of terrestrial habitats versus hydropower generation. This will be relevant for planning strategies in developing countries. These aspects could be considered in future management decisions about new hydropower projects. For example, it would be great to contrast the benefits of energy generation and the ecosystems services provided by the conservation of biodiversity.

Response – We have now expanded our Discussion to include discussion of the importance of conserving terrestrial habitats (lines 192 – 195), including their potential to harbor populations of apex predators which play a critical in ecosystem functioning. We have then contrasted that with the impacts of hydropower development (lines 195 – 200).

-In the methods section, the authors should provide additional information about the quality of datasets used for tiger and jaguar densities, especially the datasets compiled from gray literature (e.g., list the data sources and provide an assessment of their credibility). Also, a more detailed explanation is needed about how published estimates of species densities from nearby regions are used. Is any aspect of differential habitat conditions among these regions considered? Similarly, the authors need to provide the sources of information related to the list of dams (i.e., Table S1 and S2). Lastly, as I mentioned above, the assumption that tigers and jaguars would not survive the habitat loss from reservoir flooding is weak. A comprehensive analysis of the literature could resolve this along with an evaluation of evidence that the expected elevated competition under habitat reduction is feasible. However, a special consideration should be taken for populations that are already in decline due to the carrying capacity of these systems are arguably below their historical conditions so higher densities might not be an issue.

Response – We have now revised the methods section to incorporate these points. Below, we describe the changes we have made to address each point raised:

(1) List the data sources on the species densities and provide an assessment of their credibility: We have now provided the species density values and corresponding reference for each dam with which a density value has been associated. This can be found in Tables S2 and S3, for the distributions of jaguars and tigers, respectively. We also improved our explanation in the text regarding the studies used to extract density values (lines 251 – 259). For jaguars, we used a previous compilation of studies recently carried out by Jędrzejewski et al. (2018). Density estimates of jaguars were not available for some countries; for these cases we used the local estimates of jaguar density obtained by Jędrzejewski et al. (2018). For tigers, we used both published studies and unpublished reports. In addition, given the high levels of uncertainty on our estimates on the potential number of affected jaguars and tigers, we further stress that we have now changed the focus of this manuscript to the habitat loss caused by hydropower reservoirs within the distribution of jaguars and tigers.

(2) How published estimates of species densities from nearby regions are used: We have now added this explanation:

“we matched the area of each existing reservoir with the nearest available estimate of species density to obtain the potential number of affected individuals” (lines 263 – 264)

(3) Is any aspect of differential habitat conditions among these regions considered: We did not consider the different habitat conditions among the regions. While we acknowledge that this point may be relevant, we recognize that in the previous version of this manuscript we were perhaps too ambitious in estimating numbers of jaguars and tigers affected by habitat flooding with limited data. In fact, our current estimates would greatly benefit from previous analysis on the habitat suitability considering the different habitat conditions among the different regions, as well as any studies focused on the local responses of these species to habitat flooding. This would also require a standardization across the different studies reporting jaguar and tiger densities. Unfortunately, such data are not available. For this reason, we are not

able to improve our estimates of the number of individuals potentially affected by hydropower reservoirs. As such, we have re-focused this manuscript to highlight the habitat loss caused by the flooding of hydropower reservoirs across jaguar and tiger ranges.

- (4) Sources of information related to the list of dams: We have now added information on the source of the dams compiled at the country-level (Table S1).
- (5) Analysis of the literature could resolve this along with an evaluation of evidence that the expected elevated competition under habitat reduction is feasible: We have now supported the expected elevated competition for individuals displaced from flooded areas with examples from the literature (e.g., Tortato et al. 2017 and Chanchani et al. 2018) (lines 274 – 279). We have also stressed that some populations are likely already in decline due to limited carrying capacity (or other human disturbances):

“damming in relatively small forest areas already harboring reduced populations of top predators is expected to have further implications, potentially precipitating their local extinction³⁹. This might be the case for some populations of jaguars in the Atlantic Forest and Pantanal of Brazil, and for tigers in Central India (see Fig. 1).”
(lines 167 – 170)

Reviewer #2 (Remarks to the Author):

This is an interesting topic that perhaps does not receive the attention that it may deserve, although I feel that the manuscript lacks detail and depth of its analysis and interpretation of results. As I lay out in more detail, I have some caveats on omissions in the data and focus of the analysis, as well as too strong of a focus on extrapolating population effects rather than discussing habitat loss.

Response – Thank you very much for your encouraging and helpful comments. We have now revised the manuscript to include these suggestions. The main changes we have made are: (1) included previously missing data and corresponding sources for dams and predator densities, and (2) changed the focus of the manuscript to emphasize the habitat lost due to flooding following dam construction, rather than the number of potentially affected jaguars and tigers. Indeed, while estimates on the number of individuals affected by flooding were subject to

high levels of uncertainty, improving such estimates is unfeasible due to the lack of any study evaluating *in situ* jaguar/tiger responses to habitat flooding.

Line 44: The range in home range size is understated as it can be much larger for jaguars, see Morato et al. 2016 (PLoSone), McBride and Thompson 2018 (Mammalia), and Thompson et al. 2021 (Current Biology) for estimates of jaguar home range and Morato et al. 2018 (Biological Conservation) and Thompson et al. 2021 for importance of forest cover for jaguars.

Response – We thank the Reviewer for pointing out this inaccuracy. By mentioning such home range values, our purpose was simply to illustrate the very high area requirements of jaguars and tigers. To avoid confusion (see comments from Reviewer #3), we have removed this part of the text.

Lines 46-49: I think there should be a greater emphasis on the habitat loss and less on the extrapolation of population change since that is considerably more conjectural.

Response – We agree and thank the Reviewer for this important suggestion. We have now rewritten the manuscript with a focus on the habitat loss rather than on the extrapolation of population change. We believe this extensive change has resulted in a more solid manuscript.

Results: It would be helpful to have a summary table by country of number of planned projects and their status within the main text. Based upon the methods this was done but where are the results? There are no dam projects planned in tiger habitat or potential habitat in Russia or China? I think that the effects of dams on unoccupied, potential tiger habitat needs to be included. Shifting the emphasis on to habitat loss would facilitate this. I see in the methods that these areas were excluded. This is a major flaw in the analysis as it overlooks small populations or unoccupied habitat that are all important, especially for tigers. The supplement of the dam projects, needs to include an estimate of habitat loss by the construction of each. Why are just jaguars in the table and not tigers? The graphs in the supplemental material do not really convey the information well. Also, why are their graphs and results related to jaguars and dams and not for tigers? What are the results for tigers?

Response:

- (1) Summary table by country of number of planned projects and their status within the main text: We have now included this information in the main text. Instead of

including this as a table, we present this as a figure (Figure 2a-b) using the same color codes as in Figure 1 (i.e., red for existing dams and yellow for planned dams).

- (2) There are no dam projects planned in tiger habitat or potential habitat in Russia or China: Indeed, for this part of the tiger distribution, we identified only one existing dam (reservoir area = 5.1 km²) and no planned dams. We have now added this information in the text (lines 254 – 257). Because this part of the tiger distribution is not part of the tropical region, and is not expected to be significantly affected by hydropower development, we have excluded the tiger populations occurring in Russia and China (lines 257 – 259).
- (3) The effects of dams on unoccupied, potential tiger habitat needs to be included: We agree with the Reviewer that areas with very low tiger densities are also important to account for. In that sense, we revised the manuscript to additionally include existing and planned hydropower projects across the tiger range from where this species is possibly extinct.
- (4) The supplement of the dam projects needs to include an estimate of habitat loss by the construction of each. Why are just jaguars in the table and not tigers? Here we assume the Reviewer is referring to Tables S2 and S3 (previously, Tables S1 and S2), including information on each existing and planned dam identified across the jaguar and tiger ranges, respectively. In the previous version of the manuscript, we uploaded Tables S1 and S2 using a single excel file including two tables corresponding to the two species. We think that perhaps the Reviewer had only access to the table included in the first sheet of the excel document, which was the table regarding jaguars. In this revised version, we have uploaded Tables S2 and S3 in separate excel files. In addition, we previously included corresponding habitat loss for dam projects (under the column ‘reservoir area (km²)’) for every existing dam.
- (5) Why are their graphs and results related to jaguars and dams and not for tigers? What are the results for tigers? We assume the Reviewer is referring to the analysis we carried out relating losses in number of individuals and energy produced. Indeed, this analysis was only carried out for jaguars across their range in Brazil because we could only obtain information on both reservoir area and energy produced for each

existing and planned dam in Brazil. To make this more clear, we have now provided this explanation in the text (lines 80 – 82 and 293 – 296).

Although the analysis addresses habitat loss due to flooding, there are two aspects of this that are not addressed. Hydroelectric projects are generally accompanied by protected areas around their perimeters to avoid siltation. It could be argued that dams might result in a net gain of protected habitat. I am not saying that that is the case but it needs to be evaluated. This is also related to what habitat is lost, riparian habitats and flood plains are preferred habitats for tigers and jaguars, a better discussion of habitat quality in relation to loss is needed.

Response – This is an interesting point. Indeed, establishing protect areas in the reservoir surroundings does not only compensate for the environmental and social damage caused by the hydropower project, but also contributes to the increased longevity of the dam by minimizing siltation. Yet, establishing protected areas in the surroundings of hydroelectric reservoirs is not a common practice across the tropics (but see the case of Brazil, Latrubesse et al. 2017 Nature). In addition, even if protect areas are established, reservoirs constructed in previously remote areas would also become more vulnerable to habitat change in the aftermath of damming due to increased access by humans and other human disturbances (e.g., Finer and Jenkins (2012) Proliferation of Hydroelectric Dams in the Andean Amazon and Implications for Andes-Amazon Connectivity. PLoS ONE 7: e35126). In any case, the original habitat loss due to flooding is still evident, which will consequently impact existing populations of terrestrial species. This matter has now been addressed in the main text (lines 206 – 208). Furthermore, we have also considered the fact that it is possible that reservoirs flood habitat of particularly high quality for these two species, given their habitat preferences (lines 128 – 132).

Throughout the results and discussion, the authors talk in absolutes of loss of individuals and this is not the case, they are estimates and should be treated as such. Also, density estimates common with estimates of variance, the estimates of losses from dams should incorporate that uncertainty.

Response – Variance in the estimates of loss of individuals could be added by accounting for the variance of density values provided by each study. Yet, the species density values extracted from the literature are further dependent on the sampling effort and methods used. Although we could standardize the data accounting for the different sampling efforts and

sampling designs across the studies and calculate species density in the same way, we realize that this will also result in high uncertainty around the estimates on the number of individuals lost. Indeed, no study to date can provide information on what happens to the individuals potentially displaced by dams, so we could hardly make inferences on this specific topic. In that sense, we have now revised the manuscript to remove any mention of “lost” individuals, instead referring to only “potentially affected” individuals. In addition, we have also changed the focus of the manuscript to quantify habitat loss due to flooding as induced by dam construction across the jaguar and tiger ranges.

Additional effects of population and infrastructure (roads) growth associated with dams needs to be discussed. Infrastructure that allows human access negatively impacts jaguars and tigers. For example, see Carter et al. 2020 (Science Advances), Espinosa et al. 2018 (PLoSone), Thompson et al. 2021, Kerley et al. 2002 (Conservation Biology).

Response – We have now addressed the additional effects of human-related infrastructure associated with dam construction:

“First, hydropower reservoirs are increasingly located in remote areas, and their construction greatly increases human access to these frontier wilderness areas (e.g., construction of roads and transmission lines³⁴). Construction of such infrastructure contributes towards the additional loss, fragmentation and degradation of the habitat surrounding reservoirs^{10,11}. This further reduces the potential of these areas to support viable populations of jaguars^{35,36} or tigers^{31,37}, and may eventually disrupt metapopulation dynamics³⁸. Second, damming in relatively small forest areas already harboring reduced populations of top predators is expected to have further implications, potentially precipitating their local extinction³⁹” (lines 161 – 169)

Line 107: Again, talking about % habitat loss is more useful than extrapolating loss of individuals.

Response – We have now provided estimates on the habitat lost due to flooding in the results (Figure 2c-d). We have also added the percentage of habitat loss caused by habitat flooding considering the size of the geographic distributions of jaguars and tigers (lines 127).

Line 115: Although very dependent upon accurate density estimates I think this metric of individuals/kw is interesting as it puts the effects of dams into a common currency. It should

be used more. Also, per project and per country reporting in the supplemental material would be useful, and a summary within the main text as well.

Response – We are pleased with the Reviewer’s comment on our analysis on the trade-off between electricity generation and the potential number of jaguars affected, contrasting the current and future scenarios of hydropower development in Brazil. As added in the main text of the revised version of the manuscript (lines 80 – 82), we carried out this analysis with the purpose of illustrating the current and future trends in habitat loss caused by the flooding of hydropower reservoirs. We focused only on jaguars in Brazil because data on installed capacity and reservoir area for both existing and planned dams were only available for Brazil (lines 288 – 289). We have now made available the data used in this analysis, comprising information on the hydroelectric dams (including dam status, geographic coordinates, reservoir area and installed capacity), jaguar density and the estimated trade-off. This information can be found in Table S4 of the Supplementary Material.

Line 119-130: For conclusions and applications this section is weak. The implications and applications of the results need to be better framed, including what can be gained from accounting for dam effects with the conservation of these species.

Response – We have now improved the last part of the Discussion section, including information on the critical roles played by apex predators in ecosystem functioning, as flagship and umbrella species (lines 188 – 190). We have also discussed what can be gained by accounting for dam effects in terms of jaguar and tiger conservation (lines 220 – 223).

Lines 154-165: Start a new paragraph after line 165. This whole paragraph needs to be more detailed and clearer. It is not entirely clear how the data were organized and the analysis undertaken.

Response – We have now provided more details in the description of the methods. To improve clarity, we have added sub-titles across this section of the manuscript. We also added a new paragraph break, as suggested.

Line 165: Starting here an analysis of habitat loss should be included. The extrapolation to population numbers is useful for discussion but prone to a lot of uncertainty (uncertainty in density estimates, fate of displaced individuals, potential protection of buffer areas, etc.).

Response – We have now included an analysis of the habitat lost due to flooding (e.g., Figure 2). We made corresponding changes throughout the text to re-focus this manuscript to

highlight habitat loss caused by hydropower development within the distribution of jaguars and tigers.

Lines 176-177: “Only dams intersecting areas with a minimum of 0.0001 jaguars km⁻² were selected (283 dams).” What is this based upon? Why a value with so many significant digits? What is the justification for this cutoff?

Response – We considered this value so small that it would only account for an insignificant part of the home range of a jaguar, thereby not impacting the jaguar population. We have now provided the following explanation:

“Reservoirs less than 0.01 km² were considered to not significantly affect the home range of jaguars (e.g., 13.4 to 2,914.9 km²⁵³) or tigers (397 km²³¹) and were not included in further analyses.” (lines 242 – 244)

“As before, dams intersecting areas with less than 0.0001 jaguars km⁻² were considered to not meaningfully affect jaguar habitat; from a total of 294 dams, we selected 283 dams for this analysis.” (lines 291 – 293)

Lines 178-179: “We further analyzed how installed capacity of those dams predicted the loss of jaguars using a Generalized Linear Model.” Why just for jaguars? This analysis is not discussed much in the text. What was tested exactly?

Response – As indicated above, the analysis regarding the trade-off between hydroelectricity generation and potential number of affected individuals was only done for jaguars in Brazil because only for Brazil we were able to obtain data on dam installed capacity and reservoir area for both existing and planned dams. We have now re-written this part of the text to clarify what we have actually done (lines 288 – 293). In the current version of the manuscript, we did not make any GLM. Instead, we made a correlation analysis between reservoir area and installed capacity of dams, as explained in lines 296 – 298.

Reviewer #3 (Remarks to the Author):

Thank you for your submission to Nature Communications. I find your manuscript to be well written and of a topic that is most certainly under appreciated. My main comments relate to improving the framing of the problem. For instance, your Introduction does not currently

provide much information about the adverse effects resulting from hydropower. This seems to be left for the reader to implicitly understand. A short sentence or two to highlight the main impacts of hydropower would help to guide the reader on the extent of the problem. Secondly (and related), it is unclear how big a problem hydropower actually is from your Introduction (with more significant effects on tiger populations than (larger) jaguar populations), other than noting that hydropower will increase by >30% over the coming decades. Infrastructure development projects, for example, are expected to add 3 to 5 million km of road across ecosystem globally over the next 50 years (Meijer et al, 2018), with potentially dramatic consequences on ecosystem structure and function. How do impacts from hydropower compare? This is not to say that one type of land-cover change is more important to highlight than another, but it is to say that I believe a short section is needed to contextualize the rapid changes that are occurring across ecosystem globally. I consider land-cover change to be one of the greatest threats to terrestrial biodiversity. But, animals, such as jaguar still move through disturbed landscapes (see Morato et al. 2016 and 2018), up to some (although unknown) threshold of human disturbance. Dam construction and its associated conversion of terrestrial to aquatic habitat will surely make areas unusable, but what level of mortality will be caused by the loss in area and assumed decrease in landscape carrying capacity? The point here is to be careful about assumed responses or at least, to tender these consequences appropriately.

Response – We are grateful for the Reviewer’s constructive comments. We have now substantially revised the manuscript to include these suggestions. In particular, we improved the framing of the problem by including:

- (1) information on the known impacts on biodiversity following dam construction in the Introduction, which included both the impacts due to habitat flooded and additional impacts expanding beyond the area that is flooded (e.g., roads, urban and agricultural areas, all of which increase human access to previously remote sites) (2nd paragraph);
- (2) the expected results regarding the two scenarios: jaguars in the Neotropics and tigers in the Paleotropics (lines 73 – 77);
- (3) a comparison of hydropower with other land-use types; specifically, that hydropower makes the area occupied by the reservoir unusable for terrestrial species, while the areas surrounding reservoirs also tend to become highly disturbed (lines 44 – 48)
- (4) regarding the level of mortality caused by the loss in habitat area and assumed decrease in landscape carrying capacity, we have now re-focused this manuscript to highlight the habitat loss due to reservoir flooding, rather than the impacts on the

jaguar and tiger populations. Given the lack of studies on the responses of these predators to dam construction, we were not able to improve our estimates (please see above our responses to the other reviewers' similar comments). Yet, we still maintained our previous estimates of the potential number of jaguars and tigers affected by habitat flooding as that allows us to establish a comparison between the two species/scenarios of hydropower development (lines 282 – 283). Given the uncertainty in the number of jaguars and tigers affected by each reservoir, we have now added a statement to clarify this uncertainty (lines 294 – 296).

Your title does not fit the context of the manuscript. Other than the inference that current and future hydropower dams will decrease the habitat area and result in increased mortality, there is no discussion on impacts or responses of jaguar or tiger to anthropogenic factors. While this is not the focus of your analysis, what lessons are we to take from your analysis? That species area will be decreased?

Response – We have now modified the title to account for the main purpose of this study which was to account for habitat loss due to hydropower reservoirs across the range of jaguars and tigers: “*Overlooked impacts of hydropower on the habitat of jaguars and tigers*”

Further, why are jaguar and tigers chosen for the analysis? I have no problem with doing so, but some discussion of why these apex predators were chosen would be appropriate. Is it because they have large area requirements? Is it because they are sensitive to anthropogenic disturbance? Some information about the importance of apex predators would seem necessary to make your case for the importance of your findings or that these species act as umbrella species for a variety of biodiversity that will also be lost with increases in hydropower.

Response – We chose to focus on apex predator species due to their typical low densities and high area requirements, in addition to their critical role in ecosystem functioning and as umbrella species. We then chose to focus on jaguars and tigers to illustrate habitat losses in two scenarios with different history of hydropower development, the Neotropics and the Paleotropics. We have now provided an explanation underlying our choice to consider jaguars and tigers:

“Due to their low densities and large area requirements¹⁵, apex predators are expected to be particularly susceptible to habitat loss – both inside and outside the reservoir boundaries.”
(lines 53 – 55)

“These iconic apex predators play a critical role in ecosystem functioning¹⁹ and can also serve as umbrella species, enhancing the conservation of co-occurring species²⁰.” (lines 62–64)

Lastly, I noted that home range estimates provided in your Introduction range from 40–400 km². Morato et al. 2016, however, publish findings on the home range of jaguar using the Continuous Time Movement Modeling framework to more appropriately incorporate the autocorrelation structure inherent in most modern-day tracking datasets. Findings from this analysis illustrate that CTMM derived home ranges are 1–5 times larger than estimates based on previous methods, an easy update to make. I know your analysis is not based on home range estimates, but are there inaccuracies in IUCN distributions or density estimates that could result in inaccuracies of your conclusions? In the end, it would seem that your results provide a best guess approximation based on relevant assumptions, but errors do propagate and are worth further discussing.

Response – In the previous version of this manuscript, we provided values for the home range of jaguars and tigers with the purpose of illustrating the high area requirements of these species. We have now re-written the manuscript and, to avoid confusion, we have removed this part of the text. Indeed, our results do not use the size of the home range of these species, but instead rely on their geographic distribution. We used the species distribution provided by IUCN to account for habitat loss due to reservoir flooding. Although the IUCN species distribution is subject to inaccuracies for which we are not able to control, those also represent the best available data and have been widely used in the literature. Our results on the estimates of the potential number of affected individuals due to habitat flooding, however, are subject to some degree of uncertainty. Due to the lack of any *in situ* study evaluating either of these species responses to habitat flooding, we are not able to improve our estimates. Nevertheless, in this revised manuscript, we have changed the focus to emphasize our estimates of habitat loss rather than the estimates on the number of individuals. Indeed, our estimates on habitat loss were not subject to any extrapolation, as they simply account for the total area occupied by reservoirs across the geographic range of jaguars and tigers. In addition, we have added a paragraph in the Discussion to detail the study limitations,

including expectations on how some results could be biased and their potential implications (lines 159 – 183). Likewise, we urge caution when interpreting the results regarding the potential number of individuals affected by habitat flooding (lines 282 – 283 and 296 – 298).

I hope these comments are helpful and constructive in providing information towards an improved manuscript.

Reference:

Meijer, J. R., Huijbregts, M. A. J., Schotten, K. C. G. J. & Schipper, A. M. 2018. Global patterns of current and future road infrastructure. *Environ. Res. Lett.* 13.

Morato et al. 2016. Space Use and Movement of a Neotropical Top Predator: The Endangered Jaguar. *PLoS One* 11, 1–17.

Morato et al. 2018. Resource selection in an apex predator and variation in response to local landscape characteristics. *Biol. Conserv.* 228, 233–240.

Specific comments throughout the manuscript:

Abstract:

Hydropower is certainly a primary threat to freshwater biodiversity. But, I believe you should be broadening this statement considerably to lay the foundation for your manuscript. What about the catastrophic collapse on anadromous fish populations, like salmonids, where the impacts of hydropower extend well beyond freshwater systems? I would broaden this statement to showcase these immense problems. This doesn't need to be an extensive addition. You are correct that the 'primary' threat is freshwater biodiversity, but perhaps broadening this statement would better showcase the problem.

Response – We agree with this criticism and have now re-written the abstract to broaden the discussion of the impacts of hydropower as well as better fit the specific analyses made in this manuscript.

Introduction:

I think there is a need in the Introduction to focus greater attention on the impacts of hydropower and how systems have changed (reference to published sources), in an effort to guide the reader. Currently, almost no attention is provided on this aspect. How does hydropower construction compare with other forms of infrastructure development (i.e., roads, railways, pipelines) that have contributed to biodiversity decline and are increasing rapidly across terrestrial ecosystems globally. Most certainly, land-cover change is one of the greatest threats to biodiversity globally. But, how much impact does hydropower have? Perhaps this is one of the main goals of your study, with emphasis that hydropower has a much more significant effect than previous thought. This should be made clear to the reader.

Response – We have now improved the focus of our study by expanding the Introduction to include the overall impacts of hydropower on biodiversity, including a comparison with other forms of habitat change (lines 44 – 48).

Line 44 - As noted above, do you have a citation for the home ranges listed. Morato et al. 2016 calculate home ranges for jaguar across various biomes in South America using the Continuous Time Movement Modeling framework, a method that incorporates the autocorrelation structure inherent in most modern-day GPS tracking datasets. Noonan et al. (2019) - A comprehensive analysis of autocorrelation and bias in home range estimation - provide an analysis of how this method compares with other home ranges and find that previous methods underestimate home range areas. In the Morato et al. (2016) study, jaguar home ranges varied from 24 km² to 1268 km², representing a 1 - 4.8 increase in home range area from previous methods (same data, new method). Since your analysis is based specifically on distribution areas (not necessarily home ranges), you could be drastically overestimating how many individuals will be impacted by dam construction.

Response – In the previous version of this manuscript, we indicated home range values to illustrate the high area requirements of these apex predators. We have now re-written the manuscript and, to avoid confusion, we have removed the previous mentioning of the home range values in this part of the text. Regarding our estimates on the number of individuals impacted by dam construction, we agree with the Reviewer in that our estimates have high uncertainty. Indeed, those were based on species density values obtained using different sampling design and effort (even if the method for estimating species density was the same). For this reason, following the suggestion of Reviewer #2, we have now changed the focus of

our manuscript to emphasize the habitat loss caused by reservoir flooding, rather than the number of individuals potentially affected. Yet, we still maintained our previous estimates on the number of individuals affected as that allows us to compare the two species/different scenarios of hydropower development (lines 279 – 280). Nevertheless, we have now added a statement urging caution when interpreting such estimates (lines 281 – 283).

Results:

Line 62: You mention that "tigers have been hit hardest by past hydropower." Do you have a reference for this statement? One could easily argue that land-cover change resulting from high human population density is the main reason why so little habitat remains for tiger. Dams are another/next stage of land-cover change with potential additional negative effects because the "land" becomes unusable for the species.

Response – In the previous version of the manuscript, we were actually referring to our results in the way that we found a higher number of dams across tiger range (yet, a lower flooded area), when compared to that affecting the range of jaguars. To improve clarity and avoid confusion, we have now re-written this:

“The future growth of hydropower will disproportionately affect jaguar habitat” (lines 99 – 100)

Line 63: "> Ten times more dams planned in jaguar habitat." Yes, but the area is much larger. What's the percentage or density of dams on the landscape?

Response – Regarding the area flooded by planned dams, because we were not able to obtain data on the reservoir area for a considerable number of planned dams, we have only quantified the area of habitat flooded for existing dams. Following the reviewer's suggestion, we have now calculated the density of existing and planned dams across jaguar and tiger ranges (lines 98 – 89 and 108 – 110).

Line 65: The 'hotspot' cerrado. What do you mean 'hotspot'? Are you referencing the importance, biologically, of this habitat that is being rapidly lost and degraded?

Response – Indeed, we were referring to the hotspot definition sensu Myers et al. 2000. To clarify this point, we have now added the corresponding reference (line 103).

Myers, N., Mittermeier, R. A., Mittermeier, C. G., Da Fonseca, G. A., & Kent, J. (2000). Biodiversity hotspots for conservation priorities. *Nature*, 403(6772), 853-858.

Line 71: I suppose the question is whether establishing a protected area actually supports/protects these species. Far too often, protected area boundaries are altered or the protection status is changed altogether as a result of political will. Good examples exist on this topic in Kenya (i.e., Nairobi National Park boundaries altered to accommodate road construction) and Niger (i.e., the protection status of the Termit Tin Toumma Reserve - the largest terrestrial protected area in Africa - was changed to allow for oil/gas exploration). Even in the United States, various plans exist to allow for mineral extraction on protected lands. It begs the question as to the value of protecting areas if we (as a people) are not going to honor these agreements. Perhaps this is also something for the Discussion on recommendations.

Response – We have now argued in the Discussion for the appropriate management of protected areas, especially considering that reservoirs facilitate human access to previously remote areas (lines 206 – 208).

Discussion:

Line 109: "In the near future, we can expect considerable further losses of jaguars, given the elevated number of planned dams across this species range." Similar to comments made in your Methods, is there published research that you could reference on the response of either species (or related) to land-cover change/development. Surely with vast swaths of the Amazon being deforested and converted to agriculture, there must be research into how jaguar respond to these changes. Perhaps dam construction is even more deleterious because it renders the habitat unusable when terrestrial habitats are converted to aquatic systems.

Response – Indeed, any area flooded by reservoirs will become unusable for terrestrial species, including jaguars and tigers. Unfortunately, no studies to date have directly examined the fate of individuals occupying the reservoir areas prior to damming. However, based on results from some studies quantifying the responses of animals rescued from flooded habitats and released in surrounding areas, the displaced animals usually do not survive due to high competition (see Alho 2020 for a recent review) (lines 274 – 279). In this version of the manuscript, we followed the Reviewer's suggestion to expand our Discussion to include the fact that jaguar/tiger responses to habitat change will depend on prey

availability (Ramesh et al. 2013), levels of hunting pressure, and spatial requirements (Romero-Muñoz et al. 2019) (lines 171 – 173). Therefore, if the habitat surrounding reservoirs becomes degraded as previous studies have reported (Finer and Jenkins 2012, Chen et al. 2015), we would expect the density of these species to further decline. For instance, one population of marsh deer (*Blastocerus dichotomus*) in the Brazilian Pantanal declined by 54% after damming due to habitat reduction and deterioration of food availability (Andriolo et al. 2013). (lines 173 – 178).

We have also modified the previous sentence (line 109) to highlight the loss of habitat across the jaguar range, rather than the potential number of affected individuals:

“In the near future, we can expect considerable further losses in the habitat of jaguars, given the elevated number of planned dams in this region.” (lines 136 – 137)

Last paragraph/recommendations section: I think you should specifically add that scientists/conservationists need to be included in the Discussion on the potential impact of these proposed activities to provide a point-of-view about potential impact, through Environmental Impact Assessments or other, prior to actual construction.

Response – Following this suggestion, we now highlight the importance of including environmental experts to provide a comprehensive point-of-view regarding the potential impacts of damming on biodiversity and ecosystem functioning (lines 216 – 220).

Methods:

Line 155: "with the IUCN⁷ and jaguars⁸". What does that mean? Do you mean the distribution map provided by the IUCN on tiger and jaguar?

Response – Indeed, we meant the distribution maps provided by the IUCN. The issue highlighted by the Reviewer was a typo and has been now corrected.

Line 161: You provide an example of local species densities (e.g. 10)? Why are you not specifically listing your data sources? Are they too numerous to list? If so, these still should be listed in your Supplemental Materials documentation, as the data incorporated have the potential to bias your results (both positively and negatively).

Response – In the case of tigers, some of the species densities used were obtained from unpublished reports. In this revised version of the manuscript, we have now included all references used to extract species densities. To do so, we added that information to the tables

that list of dams intersecting jaguar and tiger distributions (Tables S2 and S3, respectively). In particular, we have now included the density values associated with each existing dam (used to estimate the number of individuals potentially affected due to flooding) and the corresponding reference. In addition, given the uncertainty inherent to our estimates on the number of jaguar and tiger individuals, we have now changed the emphasis of our manuscript to highlight the habitat loss caused by hydropower reservoirs intersecting jaguar and tiger ranges.

Line 170: "We assumed that the predators would not survive the habitat loss resulting from reservoir flooding; even if displaced to surrounding intact habitats, the resulting elevated competition would likely cause higher mortality and thereby maintain the estimated densities." While I agree with this assumption, is there published literature on how each species respond to habitat loss/displacement? All of your analyses are based on this assumption (loss in habitat from hydropower construction will result in a decrease in landscape carrying capacity and population decline). Important that what you are actually measuring is potential habitat loss.

Response – To date, no study has evaluated the response of jaguars or tigers to habitat loss following river damming. This limitation does not allow us to improve our estimates on the potential number of individuals “lost” due to habitat flooding following river damming. To overcome this issue and address the suggestion raised by the Reviewer, we quantified the potential habitat loss, and have now changed the emphasis of our manuscript to highlight the habitat loss caused by hydropower reservoirs intersecting jaguar and tiger ranges (e.g., Figure 2c and d). We also added a statement urging caution when interpreting our results regarding the estimates on the number of jaguar and tiger individuals potentially affected due to flooding. Indeed, these estimates allowed us to establish a comparison between the two species that occur in different scenarios of hydropower development (lines 279 – 282) or, in the case of jaguars in Brazil, to compare the current and future scenarios in the hydropower sector (lines 293 – 294).

Figures:

Fig 2. Insets c, d, and e provide little additional information to the reader. I would remove. These insets (c, d, e) are also listed in Figure 2b as 'a', 'b', and 'c' (not c,d,e)

Response – We have now removed the insets from this figure (currently Figure 3).

Fig S1. Tiger distribution (S1a) is shown, prior to jaguar distribution (S1b). In the main text, however, jaguar (1a) precede tiger (1b). It doesn't matter which is listed first, but these should be consistent. In Fig S2, tigers are again listed prior to jaguar, but in Fig S3, jaguar are once again listed first.

Response – To improve consistency between the figures and the text, we have now re-written the text to change the order in which we cite jaguars (first) and tigers (second).

Fig S3. The text should read "Number of existing (a,b) and planned (c,d) hydropower reservoirs intersection tiger (a,c) and jaguar (b,d) distributions, by country." The current way figure numbers are listed is confusing/hard to read (e.g., a,b existing).

Response – We have now modified and moved this figure to the main text (Figure 2). We have also improved clarity in the corresponding legend: *“Number of existing (red) and planned (yellow) dams intersecting **a** jaguar and **b** tiger distributions by country.”*

REVIEWERS' COMMENTS:

Reviewer #1 (Remarks to the Author):

The authors have done a good job in addressing my previous concerns. The methods are more clear and their main limitations are included in the text. Below, I have a couple of minor suggestions:

Line 23: It needs clarification about the increase in hydropower sector. Is the 30% related to number of dams? inundated areas? energy production?

Line 36: Please consider an alternative term to "hyper-biodiverse"

Line 53: Add "potential" before "impacts"

Line 93, and 137: Add "," before "but"

Line 234: Please, list the name of the local languages

Reviewer #2 (Remarks to the Author):

This manuscript was greatly improved. Apart from some comments and suggested edits made on the manuscript itself, I am comfortable with the changes made by the authors.

Reviewer #3 (Remarks to the Author):

Thank you for your thorough and thoughtful responses to all referee comments. I feel that you have done an excellent job incorporating these comments. As a result, I believe your manuscript to be much improved from its previous version, with results/conclusions that now match data you were able to collect. My only comment pertains to the fact that your methods include a section on the potential impact to predator population size. Results on this section, however, are only reported in the supplemental materials. If included in methods, these results should be reported in the manuscript, especially now that you have fully acknowledged the limitations of your analyses and highlight that they should be taken with caution. I think it would appropriate to add a sentence or two which would state something similar to "Based on the loss in area of the distribution of the each species and the individual reported density estimates, we estimate X number of individuals will be impacted by future hydropower construction, with greatest declines in population abundance in Brazil (jaguar) and India (tiger)". You could then reference Figure S2 in the main part of the manuscript and potentially report the % of the population that you expect to be affected. Your discussion would then remain unchanged, providing additional detail about these estimates, the need for further research into the (direct and importantly, indirect) impacts of dam construction on apex predator survival, and that the estimates should be taken with caution.

A few very minor other items that I noted throughout your manuscript:

Introduction (Line 65): Add 'a key' to the sentence "...hydropower expansion has been identified as 'A KEY' driver of habitat loss..." or similar.

Methods (Line 115-116): Figure should be Fig. 3a and 3b.

Discussion (Line 201-203): "The proposed measure is compatible with those recommended by other studies considering.....". These other studies should be referenced.

Discussion (Line 290): 'meaningful' should be 'meaningfully'

Response to Reviewers

REVIEWERS' COMMENTS:

Reviewer #1 (Remarks to the Author):

The authors have done a good job in addressing my previous concerns. The methods are more clear and their main limitations are included in the text. Below, I have a couple of minor suggestions:

Line 23: It needs clarification about the increase in hydropower sector. Is the 30% related to number of dams? inundated areas? energy production? Response – Changed accordingly.

Line 36: Please consider an alternative term to "hyper-biodiverse" Response – Changed accordingly.

Line 53: Add "potential" before "impacts" Response – Changed accordingly.

Line 93, and 137: Add ", " before "but" Response – Changed accordingly.

Line 234: Please, list the name of the local languages Response – Changed accordingly.

Reviewer #2 (Remarks to the Author):

This manuscript was greatly improved. Apart from some comments and suggested edits made on the manuscript itself, I am comfortable with the changes made by the authors. Response – All comments and suggestions have been incorporated in this updated version of the manuscript.

Reviewer #3 (Remarks to the Author):

Thank you for your thorough and thoughtful responses to all referee comments. I feel that you have done an excellent job incorporating these comments. As a result, I believe your manuscript to be much improved from its previous version, with results/conclusions that now match data you were able to collect. My only comment pertains to the fact that your methods include a section on the potential impact to predator population size. Results on this section, however, are only reported in the supplemental materials. If included in methods, these results should be reported in the manuscript, especially now that you have fully acknowledged the limitations of your analyses and highlight that they should be taken with caution. I think it would appropriate to add a sentence or two which would state something similar to "Based on the loss in area of the distribution of the each species and the individual reported density estimates, we estimate X number of individuals will be impacted by future hydropower construction, with greatest declines in population abundance in Brazil (jaguar) and India (tiger)". You could then reference Figure S2 in the main part of the manuscript and potentially report the % of the population that you expect to be affected. Your discussion would then remain unchanged, providing additional detail about these estimates, the need for further research into the (direct and importantly, indirect) impacts of dam construction on apex predator survival, and that the estimates should be taken with caution.

Response – We addressed this commented in lines 93 – 97. However, we only have estimates for the number of individuals potentially lost due to existing hydropower, not the number of individuals that will be potentially lost due to planned hydropower as we do not know the area to be flooded by planned dams (with the notable exception of planned dams in Brazil).

A few very minor other items that I noted throughout your manuscript:

Introduction (Line 65): Add 'a key' to the sentence "...hydropower expansion has been identified as 'A KEY' driver of habitat loss..." or similar. **Response – Changed accordingly.**

Methods (Line 115-116): Figure should be Fig. 3a and 3b. **Response – Changed accordingly.**

Discussion (Line 201-203): "The proposed measure is compatible with those recommended by other studies considering.....". These other studies should be referenced. **Response – Changed accordingly.**

Discussion (Line 290): 'meaningful' should be 'meaningfully' **Response – Changed accordingly.**